# A High-Sensitivity Fiber Biosensor Based on PVDF-Excited Surface Plasmon Resonance in the Terahertz Band

Yani Zhang [1,*] , Yiming Yao [1], Zhe Guang [2,3,*], Jia Xue [1], Qiuyang Wang [1], Jiaqin Gong [1], Zohaib Ali [2,4] and Zhongtian Yang [3]

[1] Department of Physics, School of Arts & Sciences, Shaanxi University of Science & Technology, Xi'an 710021, China; 210911040@sust.edu.cn (Y.Y.); 210911048@sust.edu.cn (J.X.); 220911033@sust.edu.cn (Q.W.); 220911031@sust.edu.cn (J.G.)

[2] School of Physics, Georgia Institute of Technology, 837 State Street NW, Atlanta, GA 30332, USA; ali.z@uaf.edu.pk

[3] Department of Biomedical Engineering, Georgia Institute of Technology, 313 Ferst Dr NW, Atlanta, GA 30332, USA; zyang669@gatech.edu

[4] Nano-Optoelectronics Research Laboratory, Department of Physics, University of Agriculture Faisalabad, Faisalabad 38040, Pakistan

[*] Correspondence: zhangyn@sust.edu.cn (Y.Z.); zguang3@gatech.edu (Z.G.)

**Abstract:** In this paper, a D-type photonic crystal fiber (PCF) with Zeonex material as the substrate and polyvinylidene fluoride (PVDF) material as the surface plasmon resonance (SPR) excitation layer is proposed for biosensing in the terahertz (THz) band. Analyzed with a finite element method, the proposed biosensor has shown excellent sensing properties for analyte refractive indices ranging from 1.32 to 1.45. With a maximum sensor resolution of $8.40 \times 10^{-7}$ refractive index unit (RIU) and a figure of merit of 39.42 RIU$^{-1}$, the maximum wavelength sensitivity and amplitude sensitivity can reach 335.00 μm/RIU and −66.01 RIU$^{-1}$, respectively. A ±2% fabrication tolerance analysis is also performed on the biosensor to prove its practical feasibility. We conclude that our proposed PCF biosensor utilizing PVDF-excited SPR can provide high sensitivity, and thus a compact, label-free, and convenient solution for biomedical liquid sensing in the THz band.

**Keywords:** biosensor; photonic crystal fiber; surface plasmon resonance; terahertz; polyvinylidene fluoride

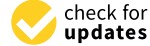



## 1. Introduction

Biosensing is a comprehensive, interdisciplinary research area across biomedicine, chemistry, physics, and electronics [1–3], which has shown great applications in not only the field of biomedicine but also other related engineering fields, such as food safety inspection and environmental monitoring [4–8]. In biosensing, surface plasmon resonance (SPR) has become a very important analysis tool, thanks to its advantages of being a fast and label-free sensing technology with high sensitivity and specificity on the analytes without any special pre-treatment of the samples [9,10]. SPR refers to the phenomenon of electron-density oscillation caused by the energy coupling between incident light and free electrons on a metallic surface: part of the energy in the incident beam is transferred to generate surface plasmon polaritons (SPPs) at corresponding resonance wavelengths [11]. SPPs are very sensitive to changes in the surrounding refractive index (RI): resonance wavelength changes with refractive index, which provides a way to detect the biomolecular contents of the measured media with high detection accuracy [12–14].

Conventional SPR sensors are usually based on prisms, utilizing a sensor configuration designed by Kretschmann, where SPR is excited using a metal film deposited on the bottom surface of a prism [15]. However, this sensor structure suffers from several disadvantages, such as its large size, high cost, difficulty in real-time measurement, and direct exposure to air that results in susceptibility to electromagnetic interference, which greatly limits

the sensor application scenarios [16]. On the other hand, the emergence of photonic crystal fibers (PCFs) resolves these problems by offering small sensor sizes and flexible and diverse design characteristics. PCF has been applied in many fields, such as bandwidth communication, thin film optics [17], fiber lasers [18], medical diagnosis [2,19], biochemical analysis [20,21], spectroscopy, and optical sensing [22,23]. Many PCF sensor properties, such as effective refractive index, confinement loss, mode area, and dispersion, can be effectively controlled by changing the air hole arrangement, pitch, and size of PCF structure parameters. Currently, PCF sensors have been widely used for analyte detection, such as gas [24], humidity [25], temperature [26,27], pH [28], acceleration [26], strain [22], and biomaterials [29,30].

PCF sensors based on SPR can be divided into two categories based on their sensing mechanisms: internal sensing and external sensing. In 2006, Hassani et al. proposed the first PCF-SPR sensor structure, which is mainly composed of two layers of air holes arranged in a regular hexagonal pattern [31]. The inner wall of the outer air holes is coated with a layer of metal film as the SPR excitation layer, and the measured medium is injected into the holes. The sensor resolution is $1.2 \times 10^{-4}$ refractive index unit (RIU), and the sensor sensitivity can reach 520 nm/RIU. In order to facilitate the phase matching between fiber core mode and SPP mode as well as to improve the sensitivity of the sensor, Rifat A. A. et al. used silver metal as the SPR excitation layer and selectively coated it on the inner wall of dielectric holes [32]. The sensitivity of this sensor can reach 2390 nm/RIU. However, these sensing structures have significant deficiencies in practical applications. For instance, achieving uniform coating of the excitation material on the inner wall of air holes and injecting analytes into the medium channels inside PCF present non-negligible challenges. Araf Shafkat proposed a gold-plated dual-core hexagonal sensing structure [33], where the gold layer is placed outside the PCF structure, which not only simplifies the detection process but also improves the fabrication feasibility. The maximum wavelength sensitivity of the sensor can reach 10,700 nm/RIU, and the maximum amplitude sensitivity can reach $1770$ RIU$^{-1}$. To facilitate the coating of metal layers on PCFs, Chao Liu et al. designed a D-type PCF where the top of the PCF cladding is polished into a D-shape, enabling the metal layer and sample to be placed on a flat top surface [34]. With this design, the metal layer is located very close to the fiber core, which favors the coupling between the fundamental mode and SPP mode to improve the sensing performance. The highest WS of this sensor can reach 15,000 nm/RIU with an SR of $6.67 \times 10^{-6}$ RIU.

Currently, research on SPPs is mainly focused on visible and near-infrared spectra. Noble metals such as gold, silver, and copper have always been the main candidate metals for generating SPPs due to their high number of conduction band electrons [35]. Terahertz (THz) refers to electromagnetic waves with frequencies in the range of 0.1–10 THz, which are located between microwave and far-infrared. Due to its unique position, the THz wave provides unique characteristics that other wavelengths cannot provide, especially in the field of biosensing. Firstly, THz radiation has low energy and does not cause damage to biological materials, making it a safe and non-invasive detection method. Moreover, THz waves have strong penetration capabilities. Compared to visible light, THz waves can penetrate non-transparent biological samples and are in line with the vibrational and rotational energy levels of biological macromolecules, thus enabling better detection of biomolecules and cells. Although noble metals, such as gold and silver, can well excite SPR on the metal surface in the visible range, they cannot excite SPR at THz due to the low energy of THz waves, which results in very small momentum coupling with electrons bound to the metal surface. Therefore, finding suitable SPR excitation materials in THz is one of the research foci in biosensing. In 2018, Jiaqi Zhu et al. achieved the coupling of two SPP modes based on graphene and PVDF and obtained an imaging sensor with a sensitivity of $730$ RIU$^{-1}$ in the THz wavelength range [36]. In 2021, Shuo Liu et al. proposed a THz surface plasmon biosensor based on MoS2 for detecting blood components. The highest WS and SR of the sensor were 715.59 µm/RIU and $1.40 \times 10^{-7}$ RIU, respectively [37].

In this work, we propose a D-shaped PCF-SPR sensor. The D-shaped fiber structure allows a close distance between the analyte and the core, which facilitates energy coupling into the SPR mode. The energy transfer from the fiber core to the SPP can be adjusted by changing the fiber pore size, thus influencing the sensing performance. PVDF material is selected as the excitation layer and coated on top of the D-shaped fiber. Detection of biological materials is achieved in the THz range for analyte refractive indices ranging from 1.32 to 1.45.

By adjusting the biosensor structural parameters, a maximum sensor resolution of $8.40 \times 10^{-7}$ RIU and a figure of merit of 39.42 RIU$^{-1}$ can be achieved, with the maximum WS and AS reaching 335.00 μm/RIU and $-66.01$ RIU$^{-1}$, respectively. A feasibility test of $\pm 2\%$ tolerance is also performed to understand the fabrication error effect on the biosensor.

## 2. Structural Design and Analysis

### 2.1. Structural Design

Figure 1 shows the cross section of the D-type PCF, containing three layers of air holes arranged in a hexagonal layout, where the innermost layer is composed of two different types of air holes: the smaller-size air hole has $d_1 = 72.5$ μm and the larger-size air hole has $d_2 = 116$ μm. When the SPR effect is excited, the transfer of energy from the core to the excitation layer in the vertical direction is facilitated due to the smaller air holes [38]. All the outer air holes have diameters of $D = 145$ μm, with air hole spacing $\Lambda = 210$ μm. The upper layer of the PCF is completely removed to form a D-shaped structure, which brings the SPP excitation material closer to the fiber core to enhance the coupling between SPP mode and fiber core mode. The polishing depth is $h = 273$ μm, which is the distance between the bottom of the PVDF layer and the fiber core. To excite the SPR effect, a PVDF layer with a thickness of $t = 12$ μm is coated on the top of the D-type fiber. Above the excitation layer is the analyte layer. A perfectly matched layer (PML) with a thickness of 1.1 times the fiber radius is set at the fiber periphery to absorb any reflected light.

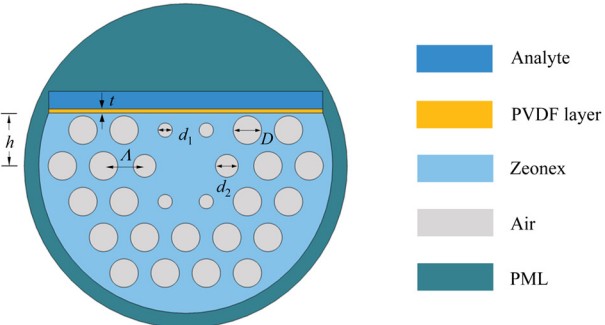

**Figure 1.** Cross-section of the proposed D-shaped PCF.

Zeonex has been selected as a substrate material for THz band fiber-optic sensors because of its unique features, such as low material absorption of 0.2 cm$^{-2}$, stable refraction index of 1.53 in the THz band, negligible material dispersion over 0.1–2.0 THz, and insensitivity to moisture or humidity [39,40].

PVDF material is a semi-crystalline ferroelectric polymer with efficient metal-like reflection and low absorption compared to metals in the terahertz range. It is also resistant to high temperatures, oxidation, and corrosion and can exhibit plasma-like Drude behavior in the THz range, making it a good choice for a dielectric layer in THz. The dielectric constant of PVDF in the THz range is defined by the Drude model as:

$$\varepsilon_{\mathrm{PVDF}}(\omega) = \varepsilon_{\mathrm{opt}} + \frac{\omega_{\mathrm{dc}} - \varepsilon_{\mathrm{opt}}}{\omega_{\mathrm{TO}}^2 - \omega^2 - \mathrm{i}\gamma\omega} \tag{1}$$

where, according to [41], $\varepsilon_{\mathrm{opt}}$ and $\omega_{\mathrm{dc}}$ are, respectively, the optical dielectric constant of the material and the low-frequency dielectric constant. $\omega$ is the angular frequency of

electromagnetic waves passing through the material. $\gamma$ is the damping frequency of the mode. $\omega_{TO}$ is the angular frequency of the transverse-optical mode of the material. In the wavelength range from 100 μm (3 THz) to 700 μm (0.43 THz), $\varepsilon_{opt}$ = 2.0, $\omega_{dc}$ = 50.0, $\omega_{TO}$ = 0.3 THz. The real part of the refractive index of PVDF in this region is less than one, while the imaginary part is positive and mostly larger than one, which is similar to the behavior of gold in the visible range. THz plasmon excitation in the air can be supported by the ferroelectric PVDF thin layer, similar to the excitation found at the metal/PVDF interface in the visible range [42].

### 2.2. Performance Analysis

The proposed sensing structure is analyzed using COMSOL Multiphysics 5.6 software, which solves the given problem using the finite element method (FEM). The FEM simulations were performed using a meshing grid with finer elements consisting of 13,351 domain elements and 1318 boundary elements. When an electromagnetic wave is incident at a specific frequency that matches the plasma wave at metal-medium interface, SPR will occur and the incident energy will be transferred to generate SPPs. Therefore, an increased confinement loss (CL) of the fundamental mode and a sharp resonance peak will appear in the loss spectrum. The resonance wavelength will also be shifted with the change in refractive index of the measured medium. The CL of the fundamental mode can be calculated by the following equation [43]:

$$\alpha = \frac{2\pi f}{c} \times 8.686 \times \text{Im}(n_{\text{eff}}) (\text{dB/m}) \tag{2}$$

where $f$ is the frequency of the incident light, $c$ is the speed of light in a vacuum, and $\text{Im}(n_{\text{eff}})$ is the imaginary part of the effective refractive index of the fundamental mode.

After obtaining the CL, the sensor sensitivity can be evaluated by two methods, namely amplitude sensitivity (AS) and wavelength sensitivity (WS), by assessing amplitude and wavelength, respectively. AS is a method to quantify the sensor sensitivity by comparing the magnitude of change in loss at a particular wavelength due to the analyte, which can be calculated by the following equation [43]:

$$S_A = -\frac{1}{\alpha(\lambda, n_a)} \frac{\delta\alpha(\lambda, n_a)}{\delta n_a} (\text{RIU}^{-1}) \tag{3}$$

where $\alpha(\lambda, n_a)$ is the CL at a particular *RI* and $\delta\alpha(\lambda, n_a)$ is the CL contrast for two adjacent analyte *RI*s.

WS is assessed by comparing the displacement of the loss peak to the sensor sensitivity, which can be calculated by the following equation [43]:

$$S_\lambda = \Delta\lambda_{\text{peak}} / \Delta n_a (\text{μm/RIU}), \tag{4}$$

where $\Delta\lambda_{\text{peak}}$ is the difference between two consecutive resonance wavelengths and $\Delta n_a$ represents the difference between two consecutive analyte *RI*s.

In addition to sensitivity, figure of merit (FOM) is also an important parameter to characterize the performance of a sensor, which can be expressed as the ratio of WS to the spectral full width at half-maximum (FWHM). FWHM refers to the width of incident wavelengths corresponding to half of the peak loss, and as the loss peak becomes sharper and more clearly identifiable, FWHM becomes smaller in the loss spectrum. The FOM can be calculated by the following equation [44]:

$$FOM = \frac{S_\lambda(\text{μm/RIU})}{FWHM(\text{μm})} \tag{5}$$

Sensor resolution (SR) is a parameter that quantifies the detection capability of a sensor. SR reflects the ability of the sensor to respond to small changes in the refractive index of

the measured medium. Assuming a minimum spectral resolution $\delta\lambda_{\min}$ and a loss peak variation $\delta\lambda_{\text{peak}}$, the SR can be obtained from the following equation [43]:

$$R_\lambda = \frac{\delta n_a \times \delta\lambda_{\min}}{\delta\lambda_{\text{peak}}} \tag{6}$$

## 3. Simulation Results and Discussion

The sensing structure proposed in this paper aims to detect biological samples. According to Choi et al. [45], by studying "human permanent normal oral keratin-forming cells (INOK) cells", "YD10 B cells", and "human oral squamous cell carcinoma (OSCC)", they concluded that for most of the normal cells the *RI* is 1.353 ± 0.008 [46,47]. So, the analyte layer *RI* values in our model are initially set to 1.36 for the parameter optimization process. The general analysis of sensing performance is first based on investigating the effect of changing geometrical parameters on the sensor sensitivity, and after obtaining the optimal combination of parameters, the sensing performance is evaluated by changing the refractive index of the analytes. The geometrical parameters studied in this paper include the excitation layer thickness $t$, polishing depth $h$, and air hole sizes $D$, $d_1$, $d_2$. Before the optimization, the initial values of the parameters are $t$ = 12 μm, $h$ = 20 μm, $D$ = 145 μm, $d_1$ = 116 μm, and $d_2$ = 72 μm.

### 3.1. Structural Optimization of the Proposed Sensing Structure

Figure 2 shows the plots of different mode fields when the refractive index of the analyte is 1.36, where the red curve is the resonance curve excited by x-polarization and the black curve is the resonance curve excited by y-polarization. It can be seen that the resonance peaks excited in the y-polarization are sharp, clearly discernible, and significantly more prominent than those in the x-polarization. This is because the TEy mode of the y-polarization has a much larger number of free electrons than the TEx mode of the x-polarization, which results in a much higher loss peak. The refractive index of the analyte does not have a significant effect on this result. In the following, we choose to continue analyzing the y-polarization effect.

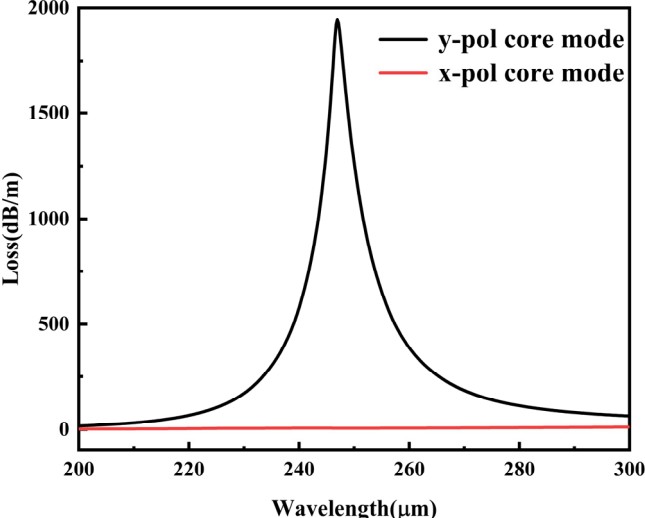

**Figure 2.** CL curves of different mode fields for analyte refractive index of 1.36.

Figure 3 shows the variations of fiber loss and effective refractive index with frequency, as shown by the red and black curves, respectively. It can be seen that in the low-frequency band, the energy is mainly confined in the fiber core, and as the frequency increases, the energy is gradually transferred from the fundamental mode to the SPP mode. The peak loss of 1941.6 dB/m appears at 246.95 μm. The effective refractive index exhibits a jump

behavior around 246.95 μm, which represents the maximum coupling strength of the core mode and SPP mode. In this scenario, the energy of the core mode rapidly transforms to the SPP mode from the fiber core region to the surface of the PVDF layer, resulting in a change in the effective refractive index. With the increase in the SPR effect, the change in the effective refractive index would be more significant. As the frequency continues to increase, the energy is transferred from the metal-dielectric interface back to the fiber core again.

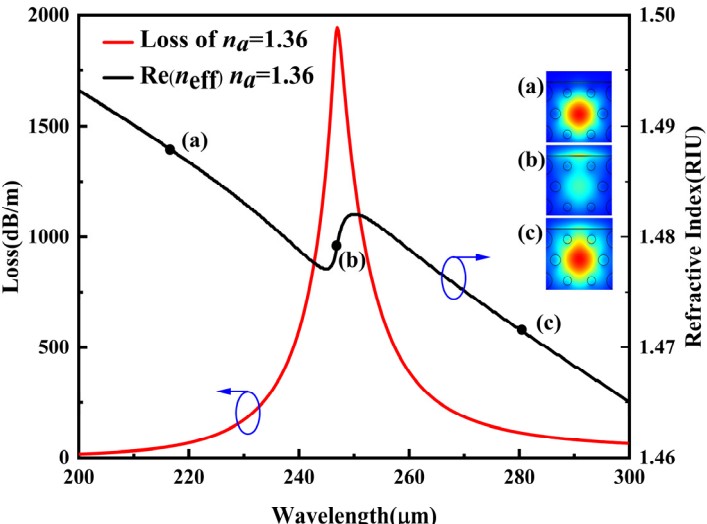

**Figure 3.** Variation in CL curve with frequency, and the electric field distributions for (**a**) core mode, (**b**) coupling mode, and (**c**) core mode.

Figure 3a–c show mode field diagrams of the fundamental mode at different frequencies. It can be seen that, for all these cases, the beam energy is well confined at the core in the proposed structure, and some energy can be transferred from the fundamental mode to the SPP mode when the phase-matching condition is satisfied, thus showing the SPR effect.

### 3.1.1. Analysis of PVDF Thickness t

The thickness of the excitation layer has a significant effect on the performance of the sensor. As Figure 4a shows, the resonance wavelength shifts to higher wavelengths as the thickness of the excitation layer increases. At excitation layer thicknesses of 5 μm, 6 μm, and 7 μm, loss peaks appear at 240.22 μm, 248.17 μm, and 257.33 μm, respectively. The loss peak values are 1475.7 dB/m, 2568.6 dB/m, and 2141.1 dB/m, respectively, with the loss peak maximum at $t = 6$ μm. This is due to a stronger coupling between the fundamental mode and SPP mode at $t = 6$ μm, compared to $t = 5$ μm, as more energy is transferred from the core to the PVDF surface. When the PVDF thickness is further increased, weaker coupling will happen due to an increased damping effect. When $n_a = 1.37$, compared to $n_a = 1.36$, the loss peaks as a whole are red-shifted. At $t = 5$ μm, the loss peak is red-shifted from 240.22 μm to 241.77 μm, and the loss peak increases from 1475.7 dB/m to 1751.2 dB/m. At $t = 6$ μm, the loss peak is red-shifted from 248.17 μm to 250.45 μm, and the loss peak decreases from 2568.6 dB/m to 2364.9 dB/m. At $t = 7$ μm, the loss peak is red-shifted from 257.33 μm to 259.79 μm, and the loss peak is reduced from 2141.1 dB/m to 1939.5 dB/m. Figure 4b shows the variations of sensor AS over wavelength at different thicknesses. As the thickness of the excitation layer changes, at $n_a = 1.36$, the highest amplitude sensitivities corresponding to $t = 5$ μm, 6 μm, and 7 μm are $-35.07$ RIU$^{-1}$, $-48.99$ RIU$^{-1}$, and $-36.05$ RIU$^{-1}$, respectively. As the highest AS is observed at $t = 6$ μm, the thickness $t = 6$ μm is chosen for further analysis.

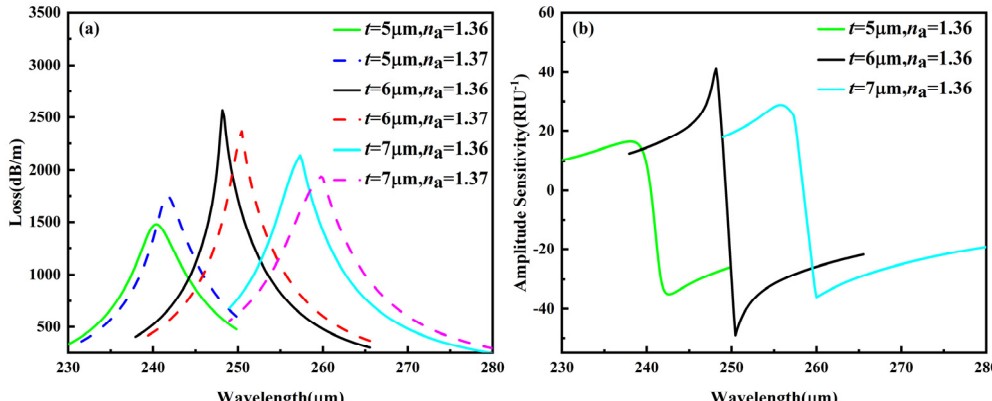

**Figure 4.** (**a**) CL curves at analyte *RI* of 1.36 (solid lines) and 1.37 (dashed lines) for *t* = 5 μm, 6 μm, and 7 μm. (**b**) AS curves at analyte *RI* of 1.36 for *t* = 5 μm, 6 μm, and 7 μm.

### 3.1.2. Analysis of Polishing Depth h

Figure 5a shows the changes in loss spectrum, and Figure 5b shows the AS over wavelength as the polishing depth is varied. The decrease in polishing depth facilitates the transfer of energy from the fiber core mode to the SPP mode. As can be seen from Figure 5a, the loss peak increases from 2414.7 dB/m to 2708.4 dB/m as the polishing depth decreases from 283 μm to 273 μm for $n_a$ = 1.36. This corresponds to loss peaks at 249.20 μm, 248.17 μm, and 247.15 μm for *h* = 283 μm, 278 μm, and 273 μm, respectively. The loss peaks move towards lower wavelengths as the polishing depth decreases. Meanwhile, as the analyte *RI* changes from $n_a$ = 1.36 to $n_a$ = 1.37, the loss peaks as a whole are red-shifted, with decreased loss values. Figure 5b shows the variations in the AS, with the highest values at *h* = 283 μm, 278 μm, and 273 μm being −45.99 RIU$^{-1}$, −48.99 RIU$^{-1}$, and −50.08 RIU$^{-1}$, respectively. In the following, we select *h* = 273 μm, as it gives the highest AS in our analysis.

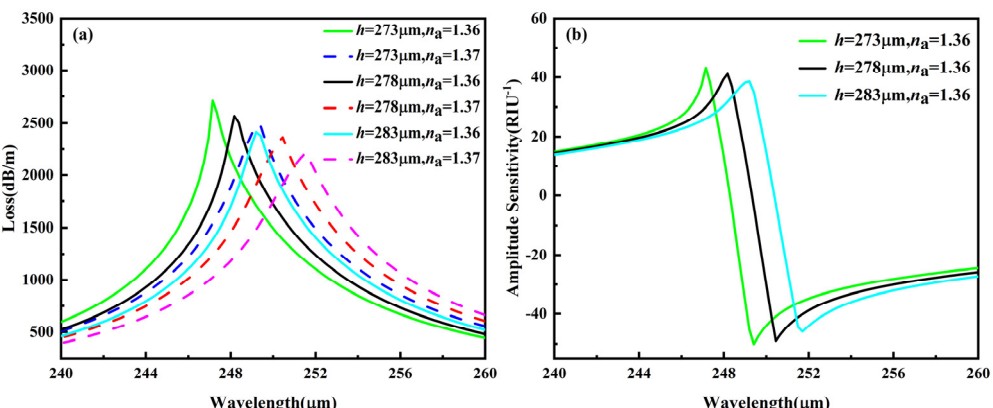

**Figure 5.** (**a**) CL curves at analyte *RI* of 1.36 (solid lines) and 1.37 (dashed lines) for *h* = 273 μm, 278 μm, and 283 μm. (**b**) AS curves at analyte *RI* of 1.36 for *h* = 273 μm, 278 μm, and 283 μm.

### 3.1.3. Analysis of Air Hole Diameter D

In the proposed fiber structure, the change in air hole size directly affects the resonance peak intensity of the fundamental mode, thus changing the AS of the sensor. The model in this paper contains three different sizes of air holes, i.e., *D*, $d_1$ and $d_2$, and the effects of outer-layer air hole diameter on the loss spectrum and AS are shown in Figure 6. Figure 6a shows the effect of the outer air hole size *D* on the loss spectrum. With the increase in the air hole size *D*, the resonance wavelength remains almost constant, while as the analyte *RI* increases from 1.36 to 1.37, the resonance peaks move towards higher wavelengths and the values of resonance peaks also increase. At *D* = 148 μm, when $n_a$ increases from 1.36 to

1.37, the loss peak value increases from 2000.6 dB/m to 2642.7 dB/m, and the resonance wavelength increases from 246.74 μm to 248.17 μm; at $D$ = 152 μm, the loss peak value increases from 2002.2 dB/m to 2683.9 dB/m, and the resonance wavelength increases from 246.95 μm to 248.58 μm; at $D$ = 156 μm, the loss peak value increases from 2013.6 dB/m to 248.79 dB/m, and the resonance wavelength increases from 247.35 μm to 248.79 μm. From Figure 6b, it can be observed that the maximum AS is $-59.89$ RIU$^{-1}$, $-65.02$ RIU$^{-1}$, and $-60.65$ RIU$^{-1}$, corresponding to $D$ = 148 μm, 152 μm, and 156 μm, respectively. Therefore, in the following analysis, we choose the air hole diameter $D$ = 152 μm.

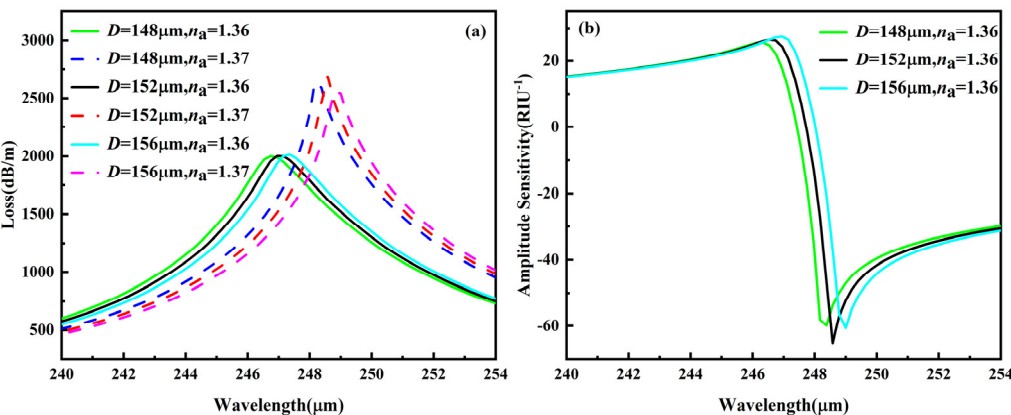

**Figure 6.** (**a**) CL curves at analyte *RI* of 1.36 (solid lines) and 1.37 (dashed lines) for $D$ = 148 μm, 152 μm, and 156 μm. (**b**) AS curves at analyte *RI* of 1.36 for $D$ = 148 μm, 152 μm, and 156 μm.

### 3.1.4. Analysis of Air Hole Diameter $d_1$ and $d_2$

Figures 7 and 8 show the effects of the inner air hole changes in the CL and AS. As the inner air hole is closer to the core mode, the changes of $d_1$ and $d_2$ have larger effects on the resonant strength of the fundamental mode. Changing the size of the inner air holes will affect the effective refractive index, which alters the ability of the core layer to confine light energy. From Figure 7a, it can be observed that, when $n_a$ = 1.36 and $d_2$ = 0.55 $D$, 0.60 $D$, and 0.65 $D$, the loss peak values are 1802.5 dB/m, 1941.6 dB/m, and 2120.3 dB/m, respectively, appearing at 246.74 μm, 246.95 μm, and 247.15 μm. The resonance peaks move to higher wavelengths with the increase in air hole sizes. When $n_a$ = 1.37 and $d_2$ = 0.55 $D$, 0.60 $D$, and 0.65 $D$, the loss peak values are 2333.6 dB/m, 2702.0 dB/m, and 2617.3 dB/m, respectively, appearing at 248.17 μm, 248.38 μm, and 248.79 μm. Figure 7b shows the variation in AS with $d_2$, which shows that the maximum AS appears at $d_2$ = 0.6 $D$, with a maximum AS value of $-66.01$ RIU$^{-1}$.

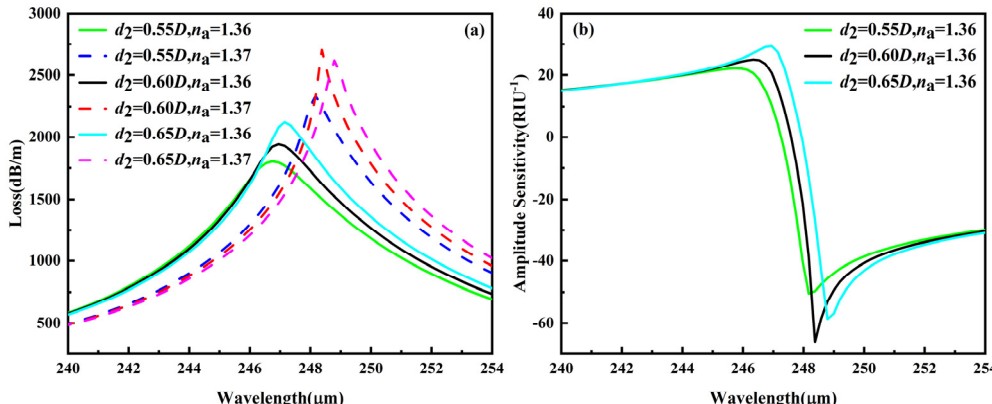

**Figure 7.** (**a**) CL curves at analyte *RI* of 1.36 (solid lines) and 1.37 (dashed lines) for $d_2$ = 0.55 $D$, 0.60 $D$, and 0.65 $D$. (**b**) AS curves at analyte *RI* of 1.36 for $d_2$ = 0.55 $D$, 0.60 $D$, and 0.65 $D$.

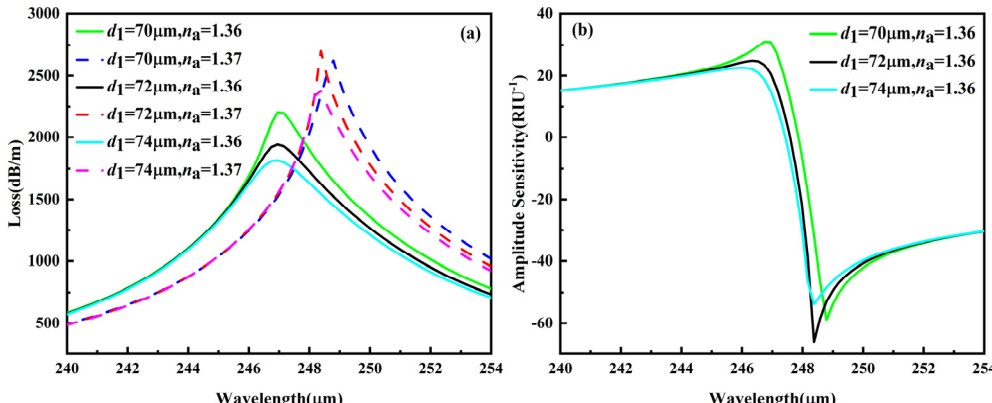

**Figure 8.** (**a**) CL curves at analyte *RI* of 1.36 (solid lines) and 1.37 (dashed lines) for $d_1$ = 70 μm, 72 μm, and 74 μm. (**b**) AS curves at analyte *RI* of 1.36 for $d_1$ = 70 μm, 72 μm, and 74 μm.

For the inner layer of small air holes, the effects of size $d_1$ change on the loss spectrum and AS are plotted in Figure 8. At $n_a$ = 1.36, when $d_1$ increases from 70 μm, 72 μm, to 74 μm, the loss peak gradually decreases, with peak values of 2203.1 dB/m, 1941.6 dB/m, and 1818.4 dB/m, respectively. However, the change in $d_1$ has little effect on the resonance wavelength, so almost no horizontal shift occurs. The resonance wavelengths all appear near 246.95 μm. At $n_a$ = 1.37, when $d_1$ is increased from 70 μm, 72 μm, to 74 μm, the loss peaks are 2625.8 dB/m, 2702.0 dB/m, and 2376.8 dB/m, respectively. When $d_1$ = 70 μm, the resonance wavelength is 248.79 μm, and when $d_1$ = 72 μm and $d_1$ = 74 μm, the resonance wavelengths are around 248.38 μm. In the following, we choose $d_1$ = 72 μm to continue our analysis, as the maximum AS appears with a value of $-66.01$ RIU$^{-1}$.

### 3.2. Sensor Performance at Optimized Parameters

Figure 9 shows the loss curves for different analyte refractive indices. It can be observed that for the analyte refractive index of 1.32–1.45, which is a very important range for biosensing, the proposed sensor has good correspondence in the loss values. Many common biological agents have *RI* values in this range, such as water (1.33), 50% sugar solution (1.42), plasma (1.35), leukocytes (1.36), hemoglobin (1.38), human intestinal mucosa (1.329–1.338), cervical cancer cells (MM231, MCF7) (1.385–1.401), and other blood components [48].

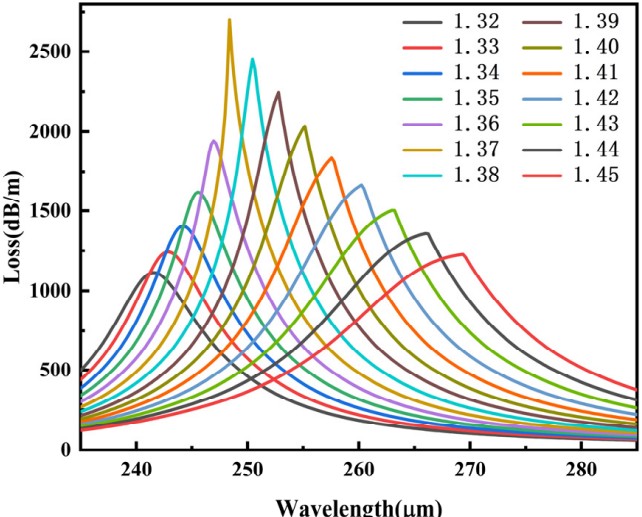

**Figure 9.** CL curves for various values of $n_a$ using optimized sensor parameters.

From Figure 9, it can be observed that as $n_a$ increases from 1.32 to 1.45 in a 0.01 increment, the resonance peaks move towards higher wavelengths. When $n_a$ increases from 1.32 to 1.37, the peak loss of the fundamental mode gradually increases, indicating that the coupling between the fundamental mode and the SPP mode gradually strengthens. Sharper peaks in the loss spectra indicate that more energy is transferred from the core to the surface of the PVDF layer. The loss peak value increases from 1113.4 dB/m at $n_a$ = 1.32 to 2701.7 dB/m at $n_a$ = 1.37, while the resonance wavelength is red-shifted from 241.57 μm to 248.38 μm. The sharper loss peak at $n_a$ = 1.37 means that the sensor has a lower FWHM, and from Equation (5), the FOM is inversely related to the FWHM, so the sensor has a higher FOM value at $n_a$ = 1.37. As $n_a$ increases from 1.37 to 1.45, the loss peak gradually decreases from 2701.7 dB/m to 1226.9 dB/m, while the resonant wavelength red-shifts from 248.38 μm to 269.36 μm with the increase in $n_a$. As $n_a$ increases, the peak loss of the fundamental mode gradually decreases, and its resonance peak gradually becomes blunt, indicating that the coupling between the fundamental mode and the SPP mode gradually becomes weaker.

Table 1 shows different sensor parameters in the range of $n_a$ = 1.32–1.45, including WS, AS, FWHM, and FOM. It can be observed that, in the range of $n_a$ = 1.32–1.37, AS increases and reaches a maximum value of −66.01 RIU$^{-1}$; when $n_a$ continues to increase to 1.45, AS gradually decreases. Compared with $n_a$ = 1.36, when $n_a$ = 1.32 or 1.44, the AS drops to a lower value, which may result in the possibility of false detection. Consequently, we set our sensor for the analyte *RI* range of 1.32–1.45. Meanwhile, as the refractive index of the analyte increases, the sensor WS tends to increase and reaches a maximum value of 335.00 μm/RIU at an analyte refractive index of 1.44. The average WS in the range of $n_a$ = 1.32–1.45 is 213.77 μm/RIU. In Equation (6), if we consider the minimum wavelength resolution of the instrument to be 0.1 nm, the sensor at $n_a$ = 1.32–1.45 for the detection of analyte *RI* changes is within the order of $10^{-7}$, and its minimum SR of $8.40 \times 10^{-7}$ is obtained at $n_a$ = 1.33, with an average SR of $5.23 \times 10^{-7}$. With Equation (5), we can obtain a maximum FOM of 39.42 for the sensor, obtained at $n_a$ = 1.37. We conclude that our proposed PCF biosensor utilizing PVDF-excited SPR can provide excellent sensing performance. A comparison of our proposed structure with the previously reported sensor is shown in Table 2.

**Table 1.** Performance analysis of the proposed biosensor with different $n_a$.

| $n_a$ | AS (RIU$^{-1}$) | WS (μm/RIU) | SR (RIU) | FWHM (μm) | FOM (RIU$^{-1}$) |
|---|---|---|---|---|---|
| 1.32 | −27.40 | 137.00 | $7.30 \times 10^{-7}$ | 12.64 | 10.84 |
| 1.33 | −29.41 | 119.00 | $8.40 \times 10^{-7}$ | 12.02 | 9.90 |
| 1.34 | −32.40 | 140.00 | $7.14 \times 10^{-7}$ | 11.23 | 12.47 |
| 1.35 | −38.38 | 142.00 | $7.04 \times 10^{-7}$ | 10.15 | 13.99 |
| 1.36 | −66.01 | 143.00 | $6.99 \times 10^{-7}$ | 8.54 | 16.74 |
| 1.37 | −51.31 | 207.00 | $4.83 \times 10^{-7}$ | 5.25 | 39.42 |
| 1.38 | −44.15 | 233.00 | $4.29 \times 10^{-7}$ | 6.94 | 33.57 |
| 1.39 | −38.15 | 236.00 | $4.24 \times 10^{-7}$ | 8.48 | 27.83 |
| 1.40 | −34.14 | 241.00 | $4.15 \times 10^{-7}$ | 10.22 | 23.58 |
| 1.41 | −31.13 | 269.00 | $3.72 \times 10^{-7}$ | 12.13 | 22.17 |
| 1.42 | −28.73 | 274.00 | $3.65 \times 10^{-7}$ | 14.10 | 19.43 |
| 1.43 | −26.71 | 303.00 | $3.30 \times 10^{-7}$ | 16.29 | 28.60 |
| 1.44 | −24.98 | 335.00 | $2.98 \times 10^{-7}$ | 18.65 | 17.96 |
| 1.45 | | | | 21.27 | |

**Table 2.** Performance comparison of the proposed and other reported PCF sensors.

| Reference | *RI* Range | Operating Wavelength (μm) | AS (RIU$^{-1}$) | WS (μm/RIU) | SR (RIU) |
|---|---|---|---|---|---|
| [42] | 1.00–1.03 | 300–460 | −44.84 | 110.00 | $2.60 \times 10^{-4}$ |
| [37] | 1.33–1.40 | 150–300 | - | 715.59 | $1.40 \times 10^{-7}$ |
| [20] | 1.33–1.40 | 0.55–0.90 | 2080.00 | 10.00 | $4.80 \times 10^{-6}$ |
| [21] | 1.35–1.40 | 0.55–1.10 | 1443.00 | 8.00 | $1.25 \times 10^{-5}$ |
| [49] | 1.30–1.40 | 0.60–1.65 | 947.00 | 22.80 | $4.38 \times 10^{-6}$ |
| This work | 1.32–1.45 | 235–285 | −66.01 | 335.00 | $8.40 \times 10^{-7}$ |

### 3.3. Analysis of Different Biomaterials

To further evaluate the performance of the sensor, the following four biomaterials are analyzed at certain refractive indices: human intestinal mucosa (1.33), white blood cells (1.36), 50% sugar solution (1.42), and breast cancer cells (1.385). As can be seen in Figure 10, the peak loss and resonance wavelengths vary for different biomaterials. Figure 10a shows the dispersion curves and loss spectra of human intestinal mucosa, where the black and red curves indicate the variations in CL and effective refractive index with wavelength, respectively. In the short wavelength range, CL is low when the energy is mainly concentrated in the core region. With the increase in wavelength, as the energy is gradually coupled from the fundamental mode to the SPP mode, the effective refractive index decreases and the CL increases. Around 242.94 μm, the refractive index changes abruptly when the fundamental mode and SPP mode meet a phase matching condition and achieve the maximum coupling, corresponding to a clearly discernible resonance peak (1242.0 dB/m) in the loss spectrum. The resonance peak indicates the coupling strength between the fundamental mode and SPP mode, and the higher the peak, the stronger the surface coupling. The FWHM corresponding to the human intestinal mucosa is 12.02 μm. Using Equations (3)–(5), we can obtain the AS, WS, and FOM of human intestinal mucosa as −29.41 RIU$^{-1}$, 119.00 μm/RIU, and 9.90 RIU$^{-1}$, respectively.

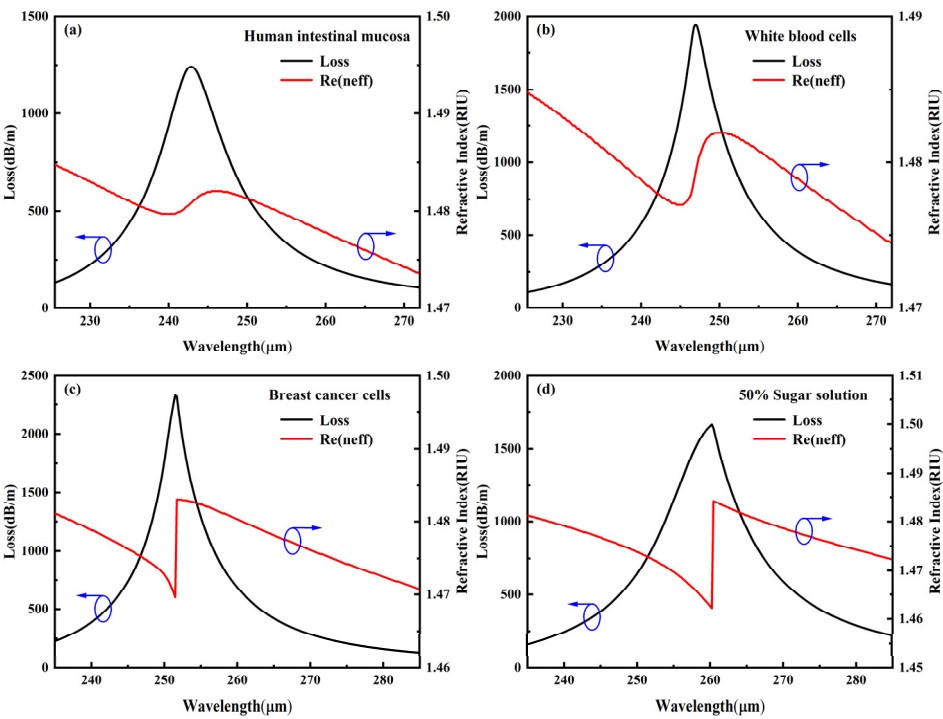

**Figure 10.** Dispersion relationship and loss spectra of (**a**) human intestinal mucosa, (**b**) white blood cells, (**c**) breast cancer cells, and (**d**) 50% sugar solution.

Similarly, we can obtain a maximum loss peak of 1941.7 dB/m at 246.95 μm for white blood cells, with AS, WS, FOM, and SR of $-66.01$ RIU$^{-1}$, 143.00 μm/RIU, 16.74 RIU$^{-1}$, and $6.99 \times 10^{-7}$, respectively.

For the breast cancer cells, the loss peak reached a maximum value of 2326.0 dB/m at 251.50 μm; the AS, WS, FOM and SR are $-41.50$ RIU$^{-1}$, 256.00 μm/RIU, 32.74 RIU$^{-1}$, and $3.91 \times 10^{-7}$, respectively.

The loss peak of 50% sugar solution reached a maximum value of 1663.6 dB/m at 260.24 μm, and the AS, WS, FOM, and SR are $-28.73$ RIU$^{-1}$, 274.00 μm/RIU, 19.43 RIU$^{-1}$, and $3.65 \times 10^{-7}$, respectively.

### 3.4. Response of Linearity

Figure 11 shows the linear fit of peak wavelengths, where the black dots are the resonance wavelengths corresponding to each na value, and the red solid line is a linearly fitted curve. The slope of the curve is 211.7076, and the linearity R-squared value is 0.9739, which indicates a high-fidelity fitting.

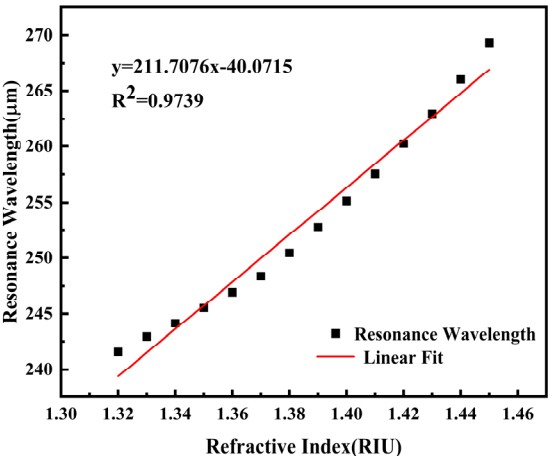

**Figure 11.** Linear fitting of the peak wavelength vs. analyte refractive index.

### 3.5. Fabrication Tolerance

In terms of fabrication, some other verified technologies are promising to realize the proposed sensing structure, such as the drawing method and 3D printing technology. According to Anthony J. et al., a microstructured polymer fiber made of Zonex has been fabricated by the drawing method [50]. The diameter of the fiber core is 400 μm, and the fiber outer diameter is 3000 μm. Similarly, using the drawing method, a Zonex polymer fiber fabricated by Woysa et al. achieved a fiber core diameter of 8.8 μm, an outer diameter of 150 μm, an average air hole diameter of 2.2 μm, and an average air hole spacing of 5.5 μm [51]. The design of photonic crystal fiber is flexible. Cordeiro et al. have discussed in detail the fabrication of microstructured optical fiber, which greatly simplifies the fabrication process of optical fibers and makes the fabrication of various optical fiber structures possible [52].

In practice, certain deviations may exist from our designed fiber sensor parameters. These deviations may have a large impact on our expected results, so it is necessary to evaluate the sensor performance over typical 1–2% fabrication tolerances [53]. Shuo Liu et al. analyzed their D-type sensor performance for 0.2–0.4% fabrication tolerances [37]. In Reference [29], the researchers performed a 2–5% fabrication tolerance analysis of their proposed sensing structure. In this paper, the fabrication tolerances of each geometric parameter in the sensor will be analyzed by changing one parameter at a time while the other geometric parameters remain unchanged. Note that the refractive index of the analyte is chosen to be 1.36.

### 3.5.1. Fabrication Tolerance for ±1–2% Variation in D

In this test, the dimension of outer air holes $D$ is varied by ±1–2%, and the results of variation are presented in Figure 12. Figure 12a presents the CL spectrum for the core mode. It can be observed that when $D$ changes from −2% to +2%, the loss peak changes from 1939.5 dB/m to 1944.9 dB/m, with an increase of only 5.4 dB/m. The resonant wavelength changes from 246.74 μm to 247.15 μm, with a deviation of less than 1 um. These indicate that a 2% change in $D$ hardly affects any energy transfer from the fundamental mode to the SPP mode, and its effect on sensor performance can be neglected. Figure 12b shows the AS spectra of $D$ in the range of 1–2%, compared to the original size. The AS increases from −66.01 $RIU^{-1}$ to −67.39 $RIU^{-1}$ with a +2% change in the size of $D$; the AS decreases to −61.65 $RIU^{-1}$ with a −2% change in the size of $D$.

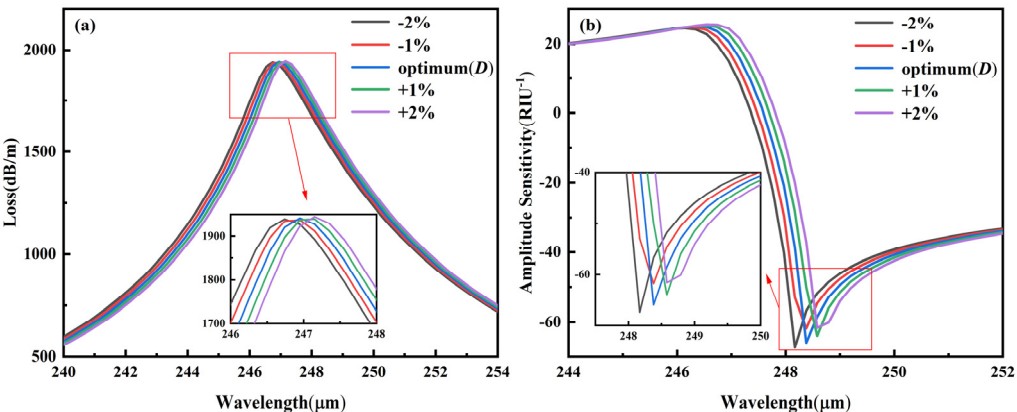

**Figure 12.** Variation in the dimension of $D$ by ±1–2%. (**a**) CL versus wavelength, and (**b**) AS versus wavelength.

### 3.5.2. Fabrication Tolerance for ±1–2% Variation in $d_1$

Figure 13a shows the CL spectrum of the core mode. The peak loss decreases from 2081.8 dB/m to 1822.2 dB/m, which is a reduction of 259.6 dB/m, as $d_1$ increases from −2% to +2%. The resonant wavelength remains constant at 246.95 μm. Compared to the large air hole size $D$, the smaller inner air hole size $d_1$ has a greater impact on the core mode energy confinement. When the internal air hole size becomes larger, the channel for energy transfer from the core mode to SPP mode becomes narrower, which causes more energy to be confined in the core and reduces the loss peak spectrum.

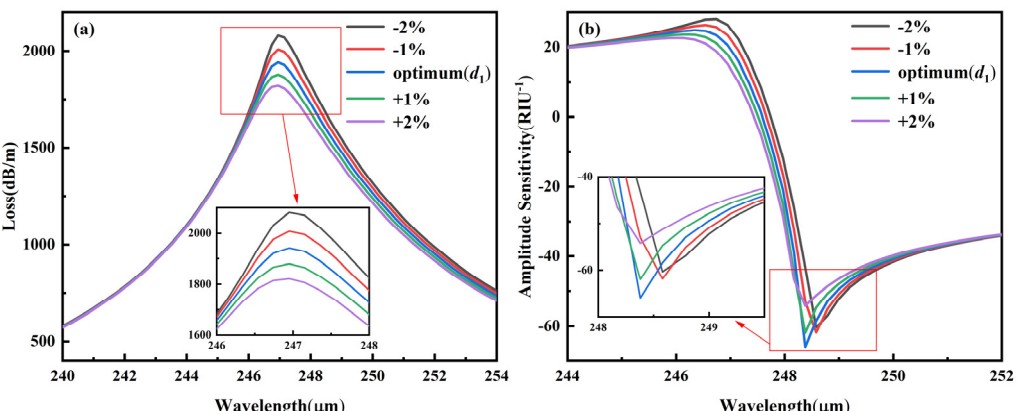

**Figure 13.** Variation in the dimension of $d_1$ by ±1–2%. (**a**) CL versus wavelength, and (**b**) AS versus wavelength.

Figure 13b shows the AS spectra of $d_1$ in the range of $\pm1$–2%, compared to the original size. The AS decreases from $-66.01$ RIU$^{-1}$ to $-60.32$ RIU$^{-1}$ with a +2% change in the size of $d_1$. The AS decreases to $-54.25$ RIU$^{-1}$ with a $-2\%$ change in the size of $d_1$.

### 3.5.3. Fabrication Tolerance for $\pm1$–2% Variation in $d_2$

Finally, we analyze the effect of the larger air hole size $d_2$ in the inner layer on the sensing performance. Figure 14a shows the CL spectrum of the core mode when $d_2$ varies in the range of $\pm1$–2%. It can be observed that its effect on both the resonance peak and resonance wavelength is not significant. When $d_2$ increases from $-2\%$ to +2%, its loss peak changes from 1905.2 dB/m to 1978.0 dB/m, which is an increase of 72.8 dB/m, while the resonant wavelengths are both 246.95 μm. Figure 14b shows the AS spectra of $d_2$ in the range of $\pm1$–2%. Compared with the original size, when the size of $d_2$ changed by $-2\%$, the AS decreased from $-66.01$ RIU$^{-1}$ to $-62.31$ RIU$^{-1}$. When the size of $d_2$ changed by $-2\%$, the AS decreased to $-62.94$ RIU$^{-1}$.

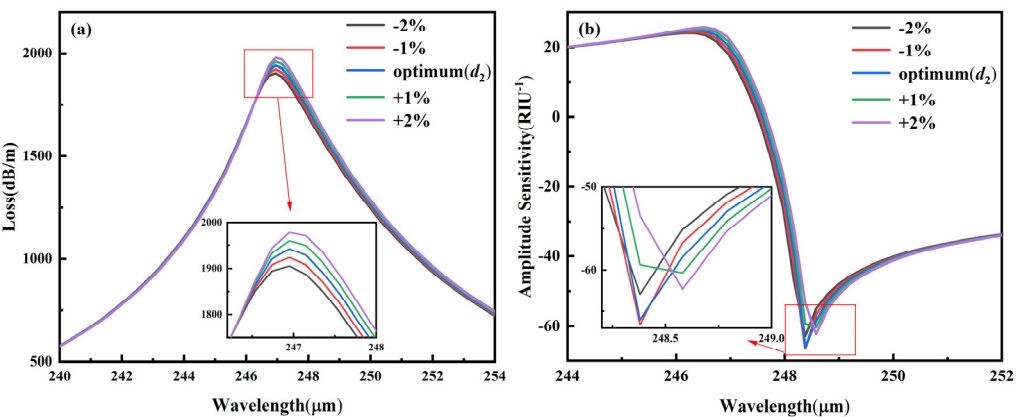

**Figure 14.** Variation in the dimension of $d_2$ by $\pm1$–2%. (**a**) CL versus wavelength, and (**b**) AS versus wavelength.

## 4. Conclusions

In this paper, a D-type terahertz SPR biosensor based on Zeonex and PVDF materials is proposed. By polishing the top of the PCF to make it D-type, the coupling strength between the fundamental mode and SPP mode is increased due to the closer distance between the fiber core and the PVDF layer. The sensing performance is investigated by optimizing parameters for the analyte refractive index in the range of 1.32–1.45. The results show that the optimal WS and AS of the sensor are 335.00 μm/RIU and $-66.01$ RIU$^{-1}$, respectively. The SR of the sensor reaches a maximum value of $8.40 \times 10^{-7}$ RIU at $n_a = 1.33$, and the highest FOM of the sensor is 39.42 RIU$^{-1}$ for $n_a = 1.37$. The sensing structure is also analyzed for fabrication tolerance in the range of $\pm2\%$, which shows that the errors are within a reasonable range and do not affect the major sensing performance. The resonance wavelength and refractive index form a clear linear relationship, with an R-squared value of 0.9739. Therefore, the designed sensor structure is considered feasible, which shows great biosensing application prospects with its wide response range, excellent sensing sensitivity, and high resolution on biological samples in the THz wavelength bands.

**Author Contributions:** Conceptualization, Y.Z. and Y.Y.; methodology, Y.Z. and Y.Y.; software, Y.Y. and J.X.; validation, Y.Z. and Y.Y.; formal analysis, Y.Z. and Y.Y.; investigation, Y.Z.; resources, Y.Y., J.X. and Z.G.; data curation, Y.Y. and Q.W.; writing—original draft preparation, Y.Y. and J.X.; writing—review and editing, Y.Z., Y.Y., Z.G. and Z.A.; visualization, Y.Y., J.X., Q.W., J.G. and Z.Y.; supervision, Y.Z., Z.G., Q.W., J.G. and Z.Y.; project administration, Y.Z. and Z.G. All authors have read and agreed to the published version of the manuscript.

**Funding:** This work was supported by Key Science and Technology Program of Shaanxi Province (No. 2021KWZ-11), China Scholarship Council (No. 202208615033), the Science and Technology project of

Xi'an City (No. 22GXFW0009), the Open Research Fund of State Key Laboratory of Transient Optics and Photonics (No. SKLST202219), Natural Science Basic Research Program of Shaanxi Province (No. 2020JQ-735), Scientific Research Plan Project of Shaanxi Education Department (No. 19JK0146), Key Science and Technology Program of Shaanxi Province (Nos. 2021QFY03-02, 2022QFY04-03), and Special Foundation of Shaanxi Provincial Department of Education (No. 20JK0543).

**Institutional Review Board Statement:** Not applicable.

**Informed Consent Statement:** Not applicable.

**Data Availability Statement:** Not applicable.

**Conflicts of Interest:** The authors declare no conflict of interest.

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
