# Peer review of "A High-Sensitivity Fiber Biosensor Based on PVDF-Excited Surface Plasmon Resonance in the Terahertz Band"

_photonics, doi:10.3390/photonics10101159_

Round 1

Reviewer 1 Report

The authors have introduced a D-type photonic crystal fiber (PCF) using Zeonex as the substrate and polyvinylidene fluoride (PVDF) as the surface plasmon resonance (SPR) excitation layer for biosensing in the terahertz (THz) band. The simulation results are promising and intriguing to the readers. However, there are several points that require attention (major revision).

1.    Please clarify what would happen if PVDF were replaced with a plasmonic metal, such as Au or Ag.

2.    Please specify whether the finite element method (FEM) simulation was conducted using proprietary code or a specific software. If software was utilized, provide the name and version for clarity. Additionally, include details about the settings used in the FEM simulations, such as mesh size and the size of PML, in the text.

3.    Equation (2) regarding the confinement loss should be provided with units.

4.    On line 157, AS and WS are introduced. On line 173, SR is mentioned. Please abbreviate these terms before their first use.

5.    In Figure 2, the plots of different mode fields are shown when the refractive index of the analyte is 1.36. It is suggested to include n=1.00 for comparison.

6.    Please explain the mechanism why the black curve in Figure 3(a) exhibits a jump behavior around 246.95 μm.

7.    Address the implications of having a refractive index (n) that is lower than 1.32 or higher than 1.45.

8.    In Figure 10(a) and 10(b), there are smooth jump curves, while Figure 10(c) and 10(d) show critical jump curves. Please elucidate the mechanism behind these differences.

9.    To enhance readers' understanding of various approaches and different frequency ranges, it is recommended to include the references of other approaches related to SPR PCF design, as presented in Photonics, 2022, 9 (12), 916, and Crystals, 2023, 13 (5), 813, in the introduction section.

10. On line 313, there is a typo, "na." Please proofread the manuscript for typos and grammar errors throughout.

11.  If possible, consider providing a comparison table to evaluate the superiority of the proposed SPR PCF in comparison to related reported works.

Reviewer 2 Report

 In this work, authors have proposed a Zeonex based D-type photonic crystal fiber with polyvinylidene fluoride material as the surface plasmon resonance excitation layer to be used for the detection of the biomaterials. The detailed analysis of structural parameters on sensing performance is discussed with explanations of underlying mechanism. The manuscript is written in a good shape logically. However, it can be considered for publication if the following comments addressed.

Question 1: Authors shows the fabrication tolerance of the proposed sensing structure, but not mention the specific manufacturing process. I highly recommend authors to propose the fabrication approach if possible.

Question 2: The study was presented as a high-sensitivity biosensor. However, this matter was only discussed with the performance of the proposed sensing structure. I suggest the authors to add the comparisons to previous sensing performance.

Question 3: Figure 12-14 is not well drawn, the scale in the inset figure is not clearly visible.

Question 4: In structural design, the description of the parameter h does not match well with what is shown in figure 1. I kindly ask the authors to better address the description of the parameter h.

Question 5: In the manuscript, the author’s abbreviations of nouns and variables is not used appropriate, such as the abbreviations ‘CL’ for confinement loss which is also used as a variable again. I suggest authors to correct it for ease of readers.

Question 6: Please further revise and polish the language expression of the whole paper in the manuscript.

Round 2

Reviewer 1 Report

My view is that the authors have fully addressed the comments now, so it can be accepted to be published in this journal.